# Structural Basis for the Inhibition of Coronaviral Main Proteases by a Benzothiazole-Based Inhibitor

**DOI:** 10.3390/v14092075

**Published:** 2022-09-18

**Authors:** Xiaohui Hu, Cheng Lin, Qin Xu, Xuelan Zhou, Pei Zeng, Peter J. McCormick, Haihai Jiang, Jian Li, Jin Zhang

**Affiliations:** 1School of Basic Medical Sciences, Nanchang University, Nanchang 330031, China; 2College of Pharmaceutical Sciences, Gannan Medical University, Ganzhou 341000, China; 3Shanghai Synchrotron Radiation Facility, Shanghai Advanced Research Institute, Chinese Academy of Sciences, Shanghai 201204, China; 4Shenzhen Crystalo Biopharmaceutical Co., Ltd., Shenzhen 518118, China; 5Jiangxi Jmerry Biopharmaceutical Co., Ltd., Ganzhou 341000, China; 6Centre for Endocrinology, William Harvey Research Institute, Barts and the London School of Medicine, Queen Mary University of London, London E1 4NS, UK

**Keywords:** coronavirus, main protease, inhibitor, warhead, YH-53

## Abstract

The ongoing spread of severe acute respiratory syndrome coronavirus-2 (SARS-CoV-2) has caused hundreds of millions of cases and millions of victims worldwide with serious consequences to global health and economies. Although many vaccines protecting against SARS-CoV-2 are currently available, constantly emerging new variants necessitate the development of alternative strategies for prevention and treatment of COVID-19. Inhibitors that target the main protease (M^pro^) of SARS-CoV-2, an essential enzyme that promotes viral maturation, represent a key class of antivirals. Here, we showed that a peptidomimetic compound with benzothiazolyl ketone as warhead, YH-53, is an effective inhibitor of SARS-CoV-2, SARS-CoV, and MERS-CoV M^pro^s. Crystal structures of M^pro^s from SARS-CoV-2, SARS-CoV, and MERS-CoV bound to the inhibitor YH-53 revealed a unique ligand-binding site, which provides new insights into the mechanism of inhibition of viral replication. A detailed analysis of these crystal structures defined the key molecular determinants required for inhibition and illustrate the binding mode of M^pro^s from other coronaviruses. In consideration of the important role of M^pro^ in developing antivirals against coronaviruses, insights derived from this study should add to the design of pan-coronaviral M^pro^ inhibitors that are safer and more effective.

## 1. Introduction

The coronavirus disease 2019 (COVID-19), caused by the severe acute respiratory syndrome coronavirus 2 (SARS-CoV-2), started in Wuhan, Hubei Province, China, in late 2019 and evolved into a worldwide pandemic with an extraordinary threat to global public health [1,2,3]. SARS-CoV-2 shares a similar genomic sequence with SARS-CoV and MERS-CoV, two viruses that caused the SARS and MERS outbreaks in 2003 and 2012 [4,5], by a sequence identity of 79% and 50%, respectively [6]. SARS-CoV-2 is the third notable coronavirus outbreak in the 21st century [7,8]. As of 7 September 2022, more than 603 million cases of COVID-19 and over 6.4 million deaths have been reported (https://covid19.who.int/, accessed on 7 September 2022), with numbers continuing to rise. The biomedical research community has made huge efforts to rapidly respond to this challenge, most notably the development of vaccines that are safe and effective against early SARS-CoV-2 strains [9,10,11]. However, the constantly emerging viral variants, including the most recently emerged Omicron sub-lineages BA.4 and BA.5, have compromised vaccine effectiveness [12,13,14,15]. Therefore, it is of utmost urgency to search for antiviral therapeutics for the prompt and effective therapy of SARS-CoV-2 infection. In this context, development of drugs that target vital enzymes of SARS-CoV-2 has been of high interest.

The main protease (M^pro^), also called 3CLpro (3C-like protease), represents an attractive antiviral drug target. After viral infection in host cells, the replicase gene encodes two large overlapping polyproteins, namely pp1a and pp1ab. M^pro^ is able to cleave pp1a and pp1ab at 11 sites, catalyzing the formation of a series of non-structural proteins critical for virus replication and transcription [16,17,18,19]. This role of M^pro^ in the viral life cycle is indispensable. Moreover, M^pro^ is highly conserved among different genera of coronaviruses and closely related homologs of M^pro^ are absent in humans [20,21]. Thus, inhibitors targeting M^pro^ can effectively impede the replication of different SARS-CoV-2 variants or future coronavirus outbreaks.

To date, various inhibitors have been developed against SARS-CoV-2 M^pro^ by using drug discovery strategies such as high-throughput screening, structure-based drug design, and drug repurposing [22,23,24,25,26,27,28]. Among these, PF-07321332 and PF-07304814 represent the most advanced inhibitors with therapeutic potential. PF-07321332 was developed by modifying the preclinical drug candidate PF-07304814, and is an analog of another peptidomimetic compound GC376, which has been used to treat cat coronavirus [27,28,29]. PF-07321332 is a potent and orally active inhibitor with good potency, safety, and pharmacokinetics properties [27]. Ritonavir can slow down the metabolic degradation of PF-07321332 in vivo when co-administrated. Unlike PF-07301332, PF-07304814 is designed to be administered by intravenous infusion [28]. Currently, Paxlovid (PF-07321332 and ritonavir) is authorized by the U.S. Food and Drug Administration (FDA) for emergency use and has been applied to the early treatment of people with mild to moderate COVID-19, while PF-07304814 has completed a phase 1b study and should soon enter a phase 2/3 study. Despite this spectacular progress, the search for SARS-CoV-2 M^pro^ inhibitors is far from over. New SARS-CoV-2 strains with mutations in the main protease are emerging [30,31]. Development of new drugs with a different mechanism of action can complement the clinical use of existing remedies to reduce the number of cases, the severity, the fatality rate, and the potential development of drug resistance. 

Previously, Thanigaimalai et al. identified a series of peptidomimetic inhibitors of SARS-CoV M^pro^ since the outbreak of SARS in 2003 [32]. In particular, YH-53 displayed not only potent SARS-CoV-2 inhibition in enzymatic and cellular antiviral assays but also a favorable absorption, distribution, metabolism, and excretion profile [33,34]. Additionally, YH-53 exhibits favorable metabolic stability, good pharmacokinetics, and has no significant toxicity [33]. All these characteristics makes YH-53 a promising lead compound in the development of anti-SARS-CoV-2 agents. Unlike PF-07301332 and PF-07304814 which carry nitrile and hydroxymethyl ketone warheads for covalent inhibition, respectively, YH-53 has a benzothiazolyl ketone as warhead. In this study, we investigated the molecular basis for YH-53 inhibiting M^pro^ using a structure-based approach. We found that YH-53 broadly inhibited main protease activities of SARS-CoV-2, SARS-CoV, and MERS-CoV. Crystal structures of the main proteases of SARS-CoV-2, SARS-CoV, and MERS-CoV bound to the inhibitor YH-53 revealed structural similarities and differences of YH-53 in binding different M^pro^s. The results reported here confirm the broad-spectrum inhibition of coronaviral main proteases by YH-53 and are highly complementary to previous studies. Thus, this study supports that YH-53 can be used as a novel scaffold and provides critical information for the optimization and design of more potent inhibitors against SARS-CoV-2.

## 2. Materials and Methods

### 2.1. Expression and Purification of M^pro^ Proteins from Human CoVs 

The full-length gene encoding M^pro^ of SARS-CoV-2, SARS-CoV, and MERS-CoV were codon-optimized, synthesized, and cloned into pET-28a vector in-frame with a His6 tag and a TEV protease cleavage site at the N terminus. Expression and purification of the His-tagged M^pro^s were performed according to a standard method described previously by our lab [21]. TEV protease was added to remove the N-terminal His tag.

### 2.2. Enzymatic Inhibition Assays 

Fluorogenic substrates as a donor and quencher pair were synthesized. The enzymatic inhibition assays were performed on a 384-well microplate according to a common protocol and PF-07321332 was used as a positive control. Briefly, YH-53 and PF-07321332 were dissolved in DMSO to produce a stock solution of 10 mM. Then, YH-53 and PF-07321332 were subjected to a 3-fold serial dilution in triplicate and mixed with a certain volume of main protease (200 nM) followed by 30 min of incubation at room temperature in the reaction buffer containing 50 mM Tris 7.3, 150 mM NaCl, and 1 mM EDTA. Subsequently, a fluorescence resonance energy transfer (FRET) substrate was added to the reaction system. The wells with only reaction buffer were used as blank controls, while the wells containing DMSO instead of tested compounds were used as vehicle treated controls. The reaction system was then monitored over a period of time and the luminescence (expressed as Relative Light Units, RLU) was recorded. By using the GraphPad Prism (v.9.3) software, inhibition activity (%) was calculated relative to vehicle (DMSO) control wells with the following formula: Inhibition(%) = 100% − [RLU(compound) − RLU(blank)/RLU(DMSO) − RLU(blank)] × 100%. Three independent experiments were performed.

### 2.3. Crystallization of M^pro^-YH-53 Complexes 

The recombinant M^pro^s were concentrated up to 5 mg/mL and incubated with YH-53 according to a 1:3 molar ratio for 30 min on ice before crystallization. Crystals were grown at 18 °C using the hanging drop vapor-diffusion method. After 3 to 5 days, the crystals of M^pro^s in complex with YH-53 were obtained. For the SARS-CoV-2 M^pro^-YH-53 complex, the final crystallization conditions were 0.1 M HEPES pH 7.5, 18% *w/v* PEG 10,000. The crystals of SARS-CoV M^pro^ in complex with YH-53 were grown in a reservoir solution comprising 0.1 M HEPES pH 7.5, 12% PEG 8000, and 10% ethylene glycol, while the crystals of MERS-CoV M^pro^ in complex with YH-53 were obtained in a solution consisting of 10% PEG 200, 0.1 M Bis-Tris-propane pH 9.0, and 20% PEG 8000.

### 2.4. Data Collection, Structure Determination, and Refinement

Before data collection, crystals were soaked in a cryoprotection solution containing 20% glycerol and then flash cooled in liquid nitrogen. Diffraction data were collected at 100 K on a macromolecular crystallography beamline 02U1 (BL02U1) at Shanghai Synchrotron Radiation Facility (SSRF, Shanghai, China). All the data sets were processed with HKL2000 software. The phase problem was solved with the molecular replacement method. Subsequent cycles of refinement to ideal resolution were performed using Phenix. The complete data collection and statistics of refinement are shown in Table 1.

## 3. Results 

### 3.1. Inhibitory Activities of YH-53 against Coronavirus M^pro^s 

M^pro^s of SARS-CoV-2, SARS-CoV, and MERS-CoV were expressed and purified as previously reported [21]. The inhibitory activities of YH-53 against M^pro^s of SARS-CoV-2, SARS-CoV, and MERS-CoV were determined by fluorescence resonance energy transfer (FRET) assay. PF-07321332 was used as a positive control with half-maximal inhibitory concentration (IC_50_) values against SARS-CoV-2 and MERS-CoV M^pro^s being 0.023 μM and 2.103 μM, respectively (Appendix A), which is similar to previous reports [27]. The results showed that YH-53 has a potent inhibition against SARS-CoV-2 M^pro^, with an IC_50_value of 0.1240 µM (Figure 1a), which is much lower than that of boceprevir but similar to that of GC376 [22]. YH-53 inhibits MERS-CoV M^pro^ with an IC_50_ value of 3.103 µM (Figure 1b). YH-53 also inhibits M^pro^ of SARS-CoV just like PF-07321332 (Appendix A). However, possibly due to the instability of SARS-CoV M^pro^ produced by our lab, the IC_50_ values of these two compounds cannot be accurately calculated. According to a previous study, YH-53 showed an IC_50_ value of 0.74 µM against SARS-CoV M^pro^ [32], which together with the present study suggests that YH-53 is a broad-spectrum inhibitor against main protease of coronaviruses. 

### 3.2. Inhibitory Mechanism of YH-53 against SARS-CoV-2 M^pro^


In order to figure out the inhibitory mechanisms of YH-53, we first determined the crystal structure of SARS-CoV-2 M^pro^ in complex with YH-53 at 1.93 Å resolution using the cocrystallization method (Table 1). The space group is *P*12_1_1 with unit cell dimensions of a = 55.56, b = 99.44, and c = 59.67 Å. The crystallographic asymmetric unit comprises a dimer form of SARS-CoV-2 M^pro^. In fact, a variety of studies previously demonstrated that only the dimeric form of M^pro^ exhibits enzymatic activity [35]. As shown in Figure 2, the M^pro^ molecule can be divided into three domains, namely, domain I (residues 10 to 99), domain II (residues 100 to 184), and domain III (residues 201 to 303). YH-53 can be found in both protomer A and protomer B (Figure 2a). Specifically, YH-53 exhibits an extended conformation in the active site, which is located in the cleft between domains I and II of SARS-CoV-2 M^pro^ (Figure 2b). YH-53 employs a unique benzothiazolyl ketone moiety as the P1′-directed warhead. The benzothiazole group of YH-53 is well accommodated in the S1′ pocket of SARS-CoV-2 M^pro^ (Figure 2b). As clearly indicated by the electron density map, the carbonyl carbon of the benzothiazole group in YH-53 forms a C-S covalent bond with the sulfur atom (2.14 Å C-S bond length) of catalytic residue Cys145 in SARS-CoV-2 M^pro^ (Figure 2c). This nucleophilic addition reaction results in the conversion of the carbonyl to a carbinol hydroxyl, which forms hydrogen-bonding interactions with the backbone NH of Cys145 and a water molecule for stabilization. An additional hydrogen-bonding interaction is formed between the nitrogen atom of the benzothiazole group and the backbone NH of Gly143 via a bridging water molecule. Additionally, several polar bonds are formed between the benzothiazole group and His41. As shown in Figure 2d,e, YH-53 contains a pyrrolidine-2-one group at the P1 position which matches the S1 pocket well. The carbonyl of the pyrrolidine-2-one group forms a hydrogen-bonding interaction with the NƐ2 atom of His163, while the nitrogen atom forms additional hydrogen-bonding interactions with the carboxy groups of Glu166 and Phe140. In addition, the carbonyl oxygen of His164 forms a hydrogen bond with the main-chain NH of the P1 moiety. YH-53 displays an isobutyl group at the P2 subsite which fits the S2 pocket well by mainly using hydrophobic interactions. Moreover, the side chain carbonyl of Gln189 forms a hydrogen-bond interaction with the main chain amide group at the P2 subsite, which creates a more closed conformation in the SARS-CoV-2 M^pro^-YH-53 complex. YH-53 presents a 4-methoxy-indole group at the P3 position, which occupies the S4 pocket of SARS-CoV-2 M^pro^. The nitrogen atom at the P3 subsite forms a hydrogen-bond interaction with the main chain carbonyl group of Glu166, while the main chain carbonyl group at the P3 subsite forms a hydrogen-bond interaction with the main chain amide group of Glu166. The methoxy group of YH-53 forms an additional hydrogen-bonding interaction with the nitrogen atom of Thr190. Therefore, YH-53 occupies the active site of SARS-CoV-2 M^pro^ by covalently binding to Cys145 and noncovalently interacting with several conserved residues, including His41, Phe140, Gly143, Cys145, His163, His164, and Glu166. This indicates the potential of YH-53 to serve as a covalent inhibitor to prevent the binding and cleavage of substrates by SARS-CoV-2 M^pro^.

Recently, the crystal structure of YH-53 in complex with SARS-CoV-2 M^pro^ has also been reported by others [33,34]. A good agreement was observed when we compared our YH-53-bound M^pro^ complex to available SARS-CoV-2 M^pro^-YH-53 structures (PDB ID 7E18 and PDB ID 7JKV) (Appendix A), with the root mean square deviation (RMSD) of equivalent Cα atomic positions between two superimposed structures being 0.396 Å and 1.441 Å, respectively.

### 3.3. Crystal Structures of YH-53 in Complex with SARS-CoV and MERS-CoV M^pro^

We also solved the crystal structures of YH-53 in complex with SARS-CoV and MERS-CoV M^pro^s at 2.04 Å and 1.99 Å resolutions (Table 1), respectively. The SARS-CoV M^pro^-YH-53 complex is crystallized in space group *P*1, while the MERS-CoV M^pro^-YH-53 complex is crystallized in space group *P*2_1_2_1_2_1_. Compared overall, the SARS-CoV M^pro^-YH-53 and MERS-CoV M^pro^-YH-53 structures show high similarity with the SARS-CoV-2 M^pro^-YH-53 structure, with the RMSD of equivalent Cɑ atoms being 1.040 Å and 1.322 Å (Figure 3a), respectively. As in the case of the SARS-CoV-2 M^pro^-YH-53 structure, ligands can be found in both protomer A and protomer B in the SARS-CoV M^pro^-YH-53 and MERS-CoV M^pro^-YH-53 structures. Two protomers in these three complexes adopt similar binding modes when bound to the ligand (Appendix A). As expected, comparison of the three complex structures shows a similar conformation of YH-53 molecule. However, there are slight differences in the orientation of YH-53 in the substrate-binding sites of M^pro^s of SARS-CoV, SARS-CoV-2, and MERS-CoV (Figure 3b). Particularly, YH-53 forms a C-S covalent bond with the catalytic cysteine residue of M^pro^s of SARS-CoV and MERS-CoV (Figure 4a,b), and forms several hydrogen-bonding interactions with some conserved residues (Appendix A), as is the case with SARS-CoV-2 M^pro^. The main differences for YH-53 interacting with M^pro^s of SARS-CoV-2, SARS-CoV, and MERS-CoV were observed at the P1′ and P3 subsites (Figure 4c,d). In the SARS-CoV-YH-53 complex, the carbinol hydroxyl at the P1′ subsite forms an additional hydrogen-bonding interaction with the backbone NH of Ser144 when compared with SARS-CoV-2 M^pro^-YH-53. Furthermore, the hydrogen bond between the methoxy group of YH-53 and Thr190 is impaired. In the case of the MERS-CoV M^pro^-YH-53 structure, the benzothiazole ring of YH-53 flips over and moves outward from the active site when compared with SARS-CoV-2 M^pro^-YH-53 and SARS-CoV M^pro^-YH-53 (Figure 3b and Figure 4d). This binding feature also helps to build a hydrogen-bonding interaction between the benzothiazole group and Ser144, but deprives the hydrogen-bonding interactions between YH-53 and residues Thr190 and Gln189. Furthermore, it is the nitrogen atom of the benzothiazole that interacts with His41, but not the sulfur atom, which forms a hydrogen bond. In addition, the main chain carbonyl group at the P2 subsite forms a hydrogen-bond interaction with a water molecule. These binding differences may contribute to the differences in inhibition observed in enzymatic assays. Therefore, these observations provide the structural basis for how YH-53 inhibits different coronavirus M^pro^s and support YH-53 as a lead compound for drug development.

## 4. Discussion

The unprecedented pandemic of SARS-CoV-2 has been threatening global health since the beginning of 2020. Thus, it is an extremely urgent task to develop effective drugs against SARS-CoV-2. Main protease of SARS-CoV-2 is highly conserved and essential to the viral life cycle, making it an appealing drug target. As SARS-CoV-2 constantly mutates, inhibitors targeting such a viral target with conserved and essential properties may improve our preparedness against future variants of concern. Moreover, residues His41 and Cys145 act as catalytic dyad, and a molecule that will interact strongly with these catalytic dyad residues may be the key to establishing strong binding to and inhibition of this enzyme. To this end, various substrate analogs carrying a chemical warhead targeting the catalytic cysteine residues have been identified as peptidomimetic inhibitors of SARS-CoV-2 M^pro^ with a covalent mechanism of action. Indeed, covalent inhibitors are of great interest as therapeutic drugs in the treatment of COVID-19. We previously reported the crystal structures of SARS-CoV-2 M^pro^ in complex with two well-designed covalent peptidomimetic drug candidates by Pfizer [36,37], namely PF-07321332 and PF-07304814. They contain a nitrile group and a hydroxymethyl ketone acting as the warheads respectively, and exhibit excellent inhibitory activity as well as potent anti-SARS-CoV-2 infection activity. 

YH-53 is a unique covalent inhibitor that utilizes the benzothiazole group as warhead. Benzothiazole-based compounds are heterocyclic systems in which a benzene ring is fused with a thiazole ring and contains nitrogen and sulfur atoms in its chemical structure. Such an electron attractive group can greatly enhance the nucleophilic reaction with catalytic residue Cys145. In this study, the enzymatic assay showed that YH-53 is a potent benzothiazole-based inhibitor against SARS-CoV-2 M^pro^. The crystal structure further revealed that YH-53 covalently binds to the catalytic cysteine of M^pro^, and forms multiple hydrogen bonds with conserved residues within the active site, which, together with previous reports, will provide additional valuable information for drug development against SARS-CoV-2 [33,34].

We also solved the crystal structures of YH-53 in complex with M^pro^s of two coronaviruses that can infect humans (MERS-CoV and SARS-CoV). Structural comparison indicates that YH-53 employs a similar binding pattern to SARS-CoV-2, SARS-CoV, and MERS-CoV M^pro^s but with some differences. These data complement the previous report about the structure of YH-53 in complex with SARS-CoV-2 M^pro^ and are beneficial to understand the broad-spectrum inhibitory activity of YH-53. Since M^pro^ is highly conserved among different coronaviruses, YH-53 can be used as a lead for the development of potent drugs against future coronaviral outbreaks.

## 5. Conclusions

There is an urgent need to develop additional drugs to expand the arsenal of antivirals against SARS-CoV-2. Main protease is an essential component in the SARS-CoV-2 life cycle and, thus, is one of the most promising targets in developing antiviral agents. We solved the crystal structures of main proteases of SARS-CoV-2, SARS-CoV, and MERS-CoV bound to the inhibitor YH-53, which contains benzothiazolyl ketone as the warhead. Detailed analysis and structural comparison provided insight into the molecular mechanism of main protease inhibition by YH-53 and could facilitate the development of broad-spectrum anti-coronavirus agents.

## Figures and Tables

**Figure 1 viruses-14-02075-f001:**
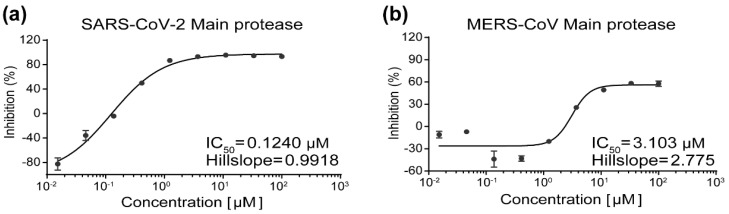
Enzymatic inhibition of YH-53 against main proteases of SARS-CoV-2 and MERS-CoV. (**a**) Inhibition of YH-53 against main protease of SARS-CoV-2. (**b**) Inhibition of YH-53 against main protease of MERS-CoV. Main proteases were preincubated in the reaction buffer with various concentrations of YH-53 at room temperature for 30 min before reacting with the FRET substrate. The IC_50_ values were calculated using the GraphPad Prism software.

**Figure 2 viruses-14-02075-f002:**
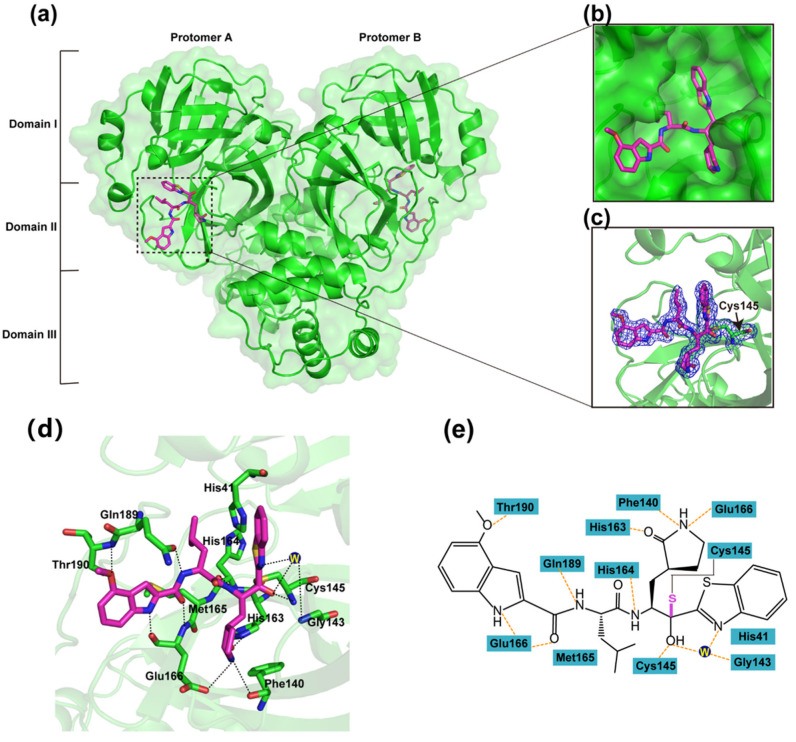
Crystal structure of SARS-CoV-2 M^pro^ in complex with YH-53. (**a**) Overall structure of SARS-CoV-2 M^pro^ in complex with YH-53. Three domains and two protomers of M^pro^ are labeled. The substrate binding pocket is situated within the black dotted box. YH-53 is shown as sticks. (**b**) An enlarged view of the substrate-binding pocket. M^pro^ is shown as surface and YH-53 is shown as sticks. (**c**) A C-S covalent bond forms between Cys145 and the benzothiazolyl ketone of YH-53. The 2*Fo–Fc* density map contoured at 1.0 σ is shown as blue mesh. (**d**) The detailed interaction in the SARS-CoV-2 M^pro^-YH-53 structure is shown with the residues of SARS-CoV-2 involved in inhibitor binding (within 3.3 Å), shown as sticks. W represents the water molecule. Hydrogen-bonding interactions are indicated as black dashed lines. (**e**) Schematic interaction between YH-53 and M^pro^. Hydrogen-bonding interactions are shown as orange dashed lines.

**Figure 3 viruses-14-02075-f003:**
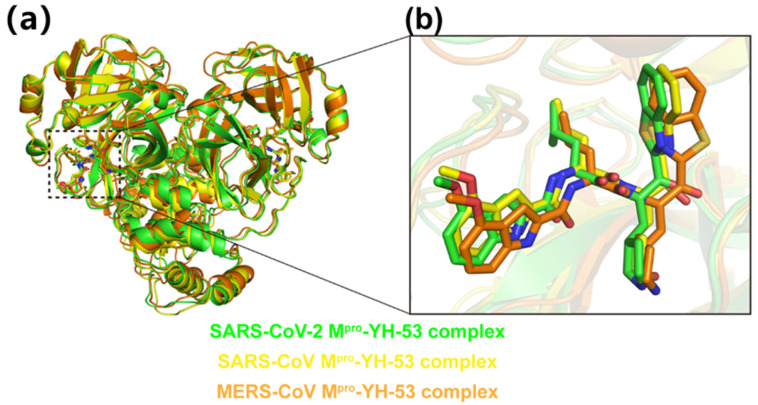
Structural comparison of M^pro^-YH-53 complexes. (**a**) Overall comparison of SARS-CoV-2 M^pro^-YH-53 (green), SARS-CoV M^pro^-YH-53 (yellow), and MERS-CoV M^pro^-YH-53 (orange) structures. YH-53 is shown as sticks. (**b**) A zoomed-in view of the substrate binding pocket of main protease. YH-53 in SARS-CoV-2 M^pro^-YH-53, SARS-CoV M^pro^-YH-53, and MERS-CoV M^pro^-YH-53 structures are shown as green, yellow, and orange sticks, respectively.

**Figure 4 viruses-14-02075-f004:**
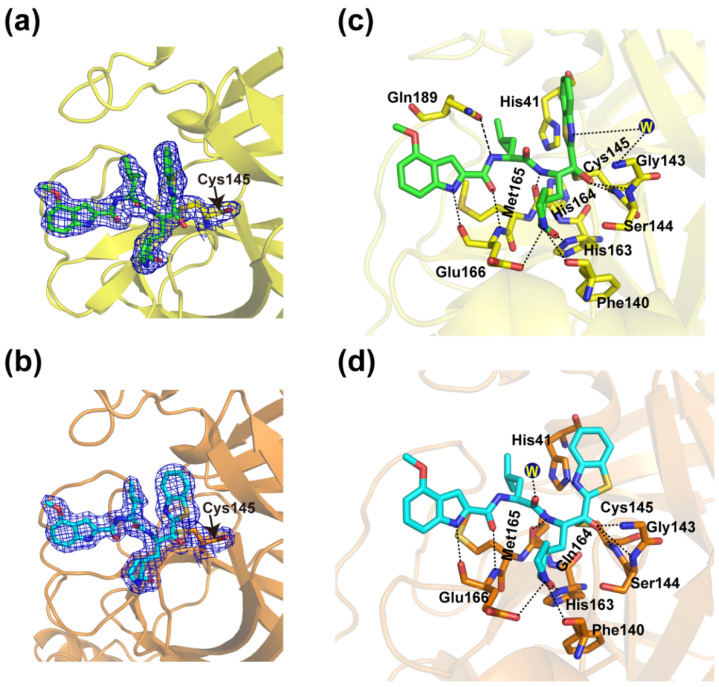
Crystal structures of SARS-CoV and MERS-CoV M^pro^s in complex with YH-53. (**a**,**b**) A C-S covalent bond forms between Cys145 of SARS-CoV M^pro^ (**a**) or MERS-CoV M^pro^ (**b**) and the benzothiazole group of YH-53. The 2*Fo-Fc* density map contoured at 1.0 σ is shown as a blue mesh. (**c**,**d**) The detailed interaction in the complex structure is shown with the residues of SARS-CoV M^pro^ (**c**) or MERS-CoV M^pro^ (**d**) involved in inhibitor binding (within 3.3 Å), displayed as sticks. Hydrogen bond interactions are shown as black dashed lines.

**Table 1 viruses-14-02075-t001:** Statistics for data processing and model refinement of M^pro^-YH-53 complexes.

	SARS-CoV-2 M^pro^-YH-53	SARS-CoV M^pro^-YH-53	MERS-CoV M^pro^-YH-53
PDB code	7XRS	7YGQ	7XRY
**Data collection**			
Synchrotron	SSRF	SSRF	SSRF
Beam line	BL02U1	BL02U1	BL02U1
Wavelength (Å)	0.97918	0.97919	0.97918
Space group	*P*12_1_1	*P*1	*P*2_1_2_1_2_1_
a,b,c (Å)	55.56, 99.44, 59.67	55.06, 60.59, 68.30	80.02, 93.8, 102.16
α,β,γ (°)	90.00, 107.93, 90.00	90.20, 120.72, 108.43	90.00, 90.00, 90.00
Total reflections	252,466	174,371	518,857
Unique reflections	46,180	50,175	53,633
Resolution (Å)	1.93(2.03–1.93)	2.04(2.15–2.04)	1.99(2.09–1.99)
R-merge (%)	5.7(49.4)	2.8(31.6)	6.2(92.6)
Mean I/σ (I)	8.1/2.5	9.3/2.5	11.0/2.6
Completeness (%)	98.9(92.9)	96.4(96.3)	99.9(99.5)
Redundancy	5.5(3.6)	3.5(3.1)	9.7(7.8)
**Refinement**			
Resolution (Å)	52.86–1.93	31.40–2.04	44.86–1.99
R_work_/R_free_(%)	19.96/24.31	20.54/23.50	22.36/25.35
Atoms	4667	4552	4800
Mean temperature factor (Å^2^)	33.9	46.8	29.0
Bond lengths (Å)	0.007	0.007	0.007
Bond angles (°)	0.952	1.034	0.92
Preferred	97.98	98.14	97.82
Allowed	2.02	1.86	2.18
Outliers	0	0	0

## Data Availability

The data presented in this study are available on request from the corresponding author.

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
