# Peer review of "Structural Basis for the Inhibition of Coronaviral Main Proteases by a Benzothiazole-Based Inhibitor"

_viruses, 2022, doi:10.3390/v14092075_

Round 1
Reviewer 1 Report
This manuscript described the structural basis of a pan-coronavirus main protease inhibitor, YH-53. The crystal structures of YH-53 complexing with Mpros from SARS-CoV-2, SARS-CoV, and MERS-CoV showed similar binding patterns and interactions mediated through conserved Mpro residues near the catalytic pocket. Enzymatic inhibition assays showed YH-53 potently inhibits the enzymatic activity of SARS-CoV-2 Mpro. However, it’s unclear how the results from the enzymatic inhibition assays correlate to virus inhibition capability. Other studies that characterize similar compounds have been published recently, which compromises the originality of this paper. In addition, here are some comments that need to be addressed:
· How conserve is the main protease among different SARS-CoV-2 variants as well as the coronavirus strains that cause common cold? More sequences can be added to Figure S4 for comparison.
· As the authors mentioned, recent studies have reported similar structures (PDB 7JKV & 7E18), please specify what are the innovations in this paper, comparing to previously published ones. In addition, the characteristics of this compound, like stability, toxicity, viral inhibition, ect. were also evaluated before, please include a brief summary of these features in the introduction.
· For enzymatic inhibition assays, please specify how %inhibition is calculated, and how many replicates were performed in each assay. In addition, it would be nice to include a positive (possibly PF-07321332) and a negative (an irrelevant compound that has no enzymatic inhibition ability) control in the assays.
Page 2 Materials and Methods 2.1: please change the heading to “Expression and purification of Mpro proteins from human coronaviruses”
Figure S1: The inhibition curve of YH-53 against SARS-CoV Mpro looks weird in the supplementary data, please provide a possible explanation here.
Figure S2: Please double-check the labels below the figures, it says GC376 in the figure caption but YH-53 in the labels. Also, please adjust the color of models in figure S2(b), it’s hard to clearly distinguish between cyan and green right now.
Reviewer 2 Report
I would like to thank for the opportunity to review this paper.
SARS-CoV-2 is a strain of coronavirus that causes COVID-19 (coronavirus disease 2019), the respiratory illness responsible for the ongoing COVID-19 pandemic. The main protease (Mpro), also called 3CLpro (3C-like protease), is an essential enzyme that promotes viral maturation, which is an attractive antiviral drug target. YH-53, a peptidomimetic compound with a unique benzothiazolyl ketone, is a potent 3CLpro inhibitor with Ki values of 6.3 nM, 34.7 nM for SARS-CoV-1 3CLpro and SARS-CoV-2 3CLpro, respectively, which was reported to strongly blocks the SARS-CoV-2 replication (ref 32, 33). And the crystal structure of SAR-CoV-2 3CL protease complex with inhibitor YH-53 (PDB ID: 7E18, 7JKV) has been reported (ref 33, 34). This paper reported the crystal structures of Mpros from SARS-CoV-2, SARS-CoV, and MERS-CoV bound to the inhibitor YH-53, which are similar to each other and also with the previously reported structures (PDB ID: 7E18, 7JKV, Figure S2). The authors also measured the IC50 of YH-53 towards Mpro of SARS-CoV-2, and MERS-CoV. Overall, this paper provides some insights, but the novelty is limited.
Of the great concern is the novelty. Besides, some other issues also need to be further addressed.
1. In Figure S1, the enzymatic inhibition of YH-53 against SARS-CoV Mpro is odd. Please further explain this curve.
2. What is the concentration of the YH-53 stocks? And what is the buffer used to dissolve it?
3. 2.3. Crystallization of Mpro-YH-53 complexes. The recombinant Mpros were concentrated up to 5 mg/mL and incubated with YH-53 according to a 3:1 molar ratio for 30 minutes. Why use the molar ratio of Mpro to YH-53 as 3:1, not 1:3?
4. The three structures reported here are similar to each other, but the space groups are different, what is the reason?
5. For the structure of MERS-CoV Mpro-YH-53 7XRY, the Rmerge is too high (92.6) for the outmost shell.
Minor Comments
No line Numbers…
Figure S2: Please check the figure legend. SARS-COV-2 Mpro-GC376?
2.1. Expression and purification of human CoVs. Should be “… of Mpros of human CoVs”?
Round 2
Reviewer 1 Report
The comments in the first draft were well addressed.
Reviewer 2 Report
I think the authors addressed my questions well and the paper is suitable for publication.